# (Almost) Free Incentivized Exploration from Decentralized Learning Agents

**Chengshuai Shi**
University of Virginia
cs7ync@virginia.edu

**Haifeng Xu**
University of Virginia
hx4ad@virginia.edu

**Wei Xiong**
The Hong Kong University of Science and Technology
wxiongae@connect.ust.hk

**Cong Shen**
University of Virginia
cong@virginia.edu

## Abstract

Incentivized exploration in multi-armed bandits (MAB) has witnessed increasing interests and many progresses in recent years, where a principal offers bonuses to agents to do explorations on her behalf. However, almost all existing studies are confined to temporary myopic agents. In this work, we break this barrier and study incentivized exploration with multiple and long-term strategic agents, who have more complicated behaviors that often appear in real-world applications. An important observation of this work is that strategic agents' intrinsic needs of learning benefit (instead of harming) the principal's explorations by providing "free pulls". Moreover, it turns out that increasing the population of agents significantly lowers the principal's burden of incentivizing. The key and somewhat surprising insight revealed from our results is that when there are sufficiently many learning agents involved, the exploration process of the principal can be (almost) free. Our main results are built upon three novel components which may be of independent interest: (1) a simple yet provably effective incentive-provision strategy; (2) a carefully crafted best arm identification algorithm for rewards aggregated under unequal confidences; (3) a high-probability finite-time lower bound of UCB algorithms. Experimental results are provided to complement the theoretical analysis.

## 1 Introduction

Multi-armed bandits (MAB) is a simple yet powerful model for sequential decision making with an exploration-exploitation tradeoff (Bubeck and Cesa-Bianchi, 2012; Lattimore and Szepesvári, 2020). In standard MAB settings, one principal, who has a long-term system-level objective, takes charge of selecting and playing arms. However, such assumption does not always hold in reality. It is often the case that arm pulls are performed by multiple different agents whose individual goals are not aligned with the system, and the principal can only observe agents' actions. One typical example is the individual buyers (agents) and the online shopping platform (the principal). Such scenarios complicate decision making and introduce significant difficulties to optimize the system performance.

Incentivized exploration has been proposed to address this problem (Frazier et al., 2014; Mansour et al., 2015). Specifically, bonuses can be offered by the principal to incentivize agents to perform specific actions, e.g., to explore their originally underrated arms. This framework provides an opportunity to reconcile different interests between the principal and agents. As a concrete example, the online shopping platform can offer discounts on certain items so that individual buyers would buy them and provide feedbacks, which can be used to optimize future strategies of the platform.

35th Conference on Neural Information Processing Systems (NeurIPS 2021).

While incentivized exploration has been investigated in the existing literature, we recognize two major limitations. First, almost all of the existing works assume the participating agents to be *myopic*, i.e., they always choose the empirically best arm. Second, it is always assumed that at each time slot, *one new agent* participates in the system, i.e., agents never stay or return. In other words, prior research mainly considers *how to incentivize one single temporary myopic agent*. These two assumptions largely limit the applicability of incentivized exploration.

In this work, we extend the study of incentivized exploration beyond the aforementioned barriers, and investigate situations with *multiple long-term strategic agents*. In particular, we focus on the scenarios (see Section 3.1) where the principal wants to identify the (overall) best arm whereas the heterogeneously involved agents only care about their different individual cumulative rewards. For such scenarios, the "Observe-then-Incentivize" (OTI) mechanism is proposed and several interesting observations are obtained. First, we find that strategic agents' intrinsic needs of learning can actually benefit principal's exploration by providing "free pulls". In other words, as opposed to myopic agents, the self interests of strategic agents can be *exploited* by the principal. Second, it turns out that increasing the number of participating agents can significantly mitigate the principal's burden on incentivizing, which highlights the importance of increasing the population of agents. A crux of our findings is the following intriguing conceptual message: when there are sufficiently many learning agents involved, the exploration process of the principal could be (almost) free.

Behind these findings, three novel technical components play critical roles in the design and analysis of OTI, all of which may have independent values.

- First, a simple yet provably effective incentive-provision strategy is developed, which can efficiently regulate strategic agents' behaviors and serves as the foundation of the algorithm analysis.
- Second, a best arm identification algorithm is carefully crafted to tackle the varying amounts of local information from heterogeneous agents. This setting itself is novel in best arm identification.
- A high-probability finite-time lower bound of UCB algorithms (Auer et al., 2002) is proved, which contributes to a better understanding of the celebrated UCB.

These insights and techniques are unique in incentivizing multiple long-term strategic agents, which may find applications in related problems, and encourage future research in this direction.

## 2   Related Works

**Incentivized exploration.** Since proposed by Frazier et al. (2014); Kremer et al. (2014), many progresses have been made in incentivized exploration in MAB. Especially, there exist two lines of studies. The first one (Kremer et al., 2014; Mansour et al., 2015, 2016; Immorlica et al., 2020; Sellke and Slivkins, 2020) assumes the principal can observe the full history while the agents cannot, and the principal leverage such information advantage to perform incentivizing. The second line, which our setting follows, considers a publicly available history while the incentives are done through compensations. This idea is first introduced by Frazier et al. (2014) and generalized by Han et al. (2015), both on Bayesian settings. The non-Bayesian case, as adopted in this work, is first studied by Wang and Huang (2018), and recently extended by Liu et al. (2020); Wang et al. (2021).

However, the aforementioned works mainly consider that one new myopic agent enters the system at each time slot and leaves afterward. The only exception is Mansour et al. (2016), where multiple but still temporary and myopic agents are considered. This work differs from them in considering *multiple long-term strategic agents*. In addition, another notable difference is that almost all prior works focus on regret minimization for the principal, instead of best arm identification.

One important related work is Chen et al. (2018), which studies temporary myopic agents with heterogeneous preferences. Free explorations are also observed there because heterogeneous preferences result in agents exploiting all arms. However, the "free pulls" of OTI is provided by strategic agents' intrinsic needs for explorations, which is fundamentally different. Regardless of the differences, both results show the value of further investigating agents' behaviors in incentivized exploration.

**Federated MAB.** This work is related to and can potentially contribute to the studies of federated MAB (FMAB) (Shi and Shen, 2021; Shi et al., 2021; Zhu et al., 2021), which considers a similar framework of multiple heterogeneous agents and a global principal. These studies assume agents would unconditionally give up learning their own local ones and naively follow global instructions. which seldom holds in practice and makes those approaches unrobust. However, the proposed

incentivizing exploration scheme achieves the "best" of both worlds, i.e., agents learn local models with additional compensations and the principal learn the global model via a small amount of cost.

# 3 Incentivized Exploration from Decentralized Learning Agents

## 3.1 Motivation

Consider the following scenario: one company (the principal) can manufacture several products, and it would like to identify one product that best suits the market. A natural strategy is to perform a market survey by having a group of users to try these products for some time and observing their feedbacks. However, different users have different preferences over these products and they are also learning in this process. Once they identify their preferred products, there is little incentive for them to explore others, which limits the information gathering by the company. Such a scenario is common in real life. For example, telecommunications companies such as AT&T (the principal) try to find the optimal channel to serve the clients (agents) of an area through a period of trials; content-providing websites such as YouTube (the principal) test the proposed features with prospective users (agents).

The idea of incentivized exploration fits in these practical scenarios perfectly since it can be utilized to provide extra bonuses for users to try different products. However, the principal now faces the problem of *how to incentivize multiple long-term strategic agents simultaneously*, which has not been investigated in the prior research to the best of our knowledge. In addition, similar settings of applications are also considered in the recently proposed federated MAB (FMAB) (Shi and Shen, 2021; Zhu et al., 2021). However, current FMAB studies only consider naive users who always follow the principal's instructions, and cannot be applied to strategic users that have individual learning abilities and objectives.

## 3.2 Agents and the Principal

Following the motivation example in Section 3.1, we consider a total of $M$ available decentralized agents, each of which interacts with a local bandit environment equipped with the same set of $K$ arms (referred to as *local* arms) but with agent-dependent arm utilities. Namely, at time step $t$, a reward $X_{k,m}(t) \in [0,1]$ is associated with agent $m$'s action of pulling arm $k$, which is sampled independently from an unknown distribution whose expectation is $\mu_{k,m} := \mathbb{E}[X_{k,m}(t)]$. In general, $\mu_{k,m} \neq \mu_{k,n}$ for $m \neq n$. These agents are (naturally) assumed to be decentralized and self-interested, i.e., their goal is to collect as much (of their own) reward as possible during a certain time horizon.

Besides these heterogeneous agents, there is also a *principal*. The principal faces a global bandit game, which also has the same set of $K$ arms (referred to as *global* arms for distinction). The expected rewards of global arm $k$ are the exact average of corresponding local arms, i.e., $\mu_k := \frac{1}{M} \sum_{m \in [M]} \mu_{k,m}$. The principal's goal is to identify the optimal global arm with a certain confidence $1 - \delta$ within a time horizon $T$, where $\delta \in (0,1)$ is a pre-fixed constant of the failure probability.

Without loss of generality, each local bandit game is assumed to have one unique optimal local arm, which is denoted as $k_{*,m} := \arg\max_{k \in [K]} \mu_{k,m}$ for agent $m$ and its expected reward is $\mu_{*,m} := \mu_{k_{*,m},m}$. Correspondingly, the sub-optimal gap for arm $k \neq k_{*,m}$ is defined as $\Delta_{k,m} := \mu_{*,m} - \mu_{k,m}$ and for arm $k_{*,m}$, we denote $\Delta_{k_{*,m},m} = \Delta_{\min,m} := \min_{k \neq k_{*,m}} \Delta_k$. Similarly, we also assume there is one unique optimal global arm $k_* := \arg\max_{k \in [K]} \mu_k$ in the global bandit game, whose expected reward is $\mu_* := \mu_{k_*}$. The global sub-optimal gap for arm $k \neq k_*$ is defined as $\Delta_k := \mu_* - \mu_k$ and for arm $k_*$, we denote $\Delta_{k_*} = \Delta_{\min} := \min_{k \neq k_*} \Delta_k$. In addition, the time horizon $T$ is assumed to be known to the principal but not to agents, as it is often the principal who determines such a horizon.

## 3.3 Interaction and Observation Model

While the agents can directly pull their local arms and observe rewards, the principal cannot interact with the global game. Instead, she can only observe local actions and rewards. In other words, with agent $m$ pulling arm $k$ at time $t$ and getting reward $X_{k,m}(t)$, the principal can also observe the action $k$ and the reward $X_{k,m}(t)$, which may be used to estimate expected rewards of global arms. It is helpful to interpret the global game as a virtual global characterization of all the local games (although it does not necessarily align with any specific one), which results in the challenge that it cannot be directly interacted with but can only be inferred indirectly.

This indirect information gathering introduces significant difficulties for the principal. In particular, it is challenging to obtain sufficient local information from self-interested agents to aggregate precise global information. For example, an arm $k$ can be exactly, or very close to, the optimal one in the global model, but also be highly sub-optimal on agent $m$'s local model. Thus, the principal needs large amount of local explorations on arm $k$ to estimate it, which contradicts with agent $m$'s willingness as arm $k$ may not provide high local rewards. Thus, the task of best (global) arm identification is likely to fail only with this passive gathering of information, which is numerically verified in Section 6.

### 3.4 Incentivized Exploration

To address these challenges, we resort to the paradigm of *incentivized exploration*, which provides a means for the principal to influence local actions. At time step $t$, the principal can provide extra bonuses for agents to explore arms, which will be announced to the agents before their decision making. Specifically, at time $t$, the bonus on arm $k$ for agent $m$ is denoted as $I_{k,m}(t) \geq 0$. If agent $m$ chooses arm $k$, she has an observation of $X_{k,m}(t)$ but obtains a reward $X'_{k,m}(t) = X_{k,m}(t) + I_{k,m}(t)$. Intuitively, if the principal wants to incentivize agent $m$ to explore a certain arm $k$ against the agent's original willingness, she should give a high bonus, i.e., $I_{k,m}(t) > 0$; otherwise, she should spare no bonus, i.e., $I_{k,m}(t) = 0$. In this way, the principal can leverage the extra bonuses to have agents gather her desired local information. The interaction model with incentivized exploration between the principal and one agent (among overall $M$ agents) is illustrated in Fig 1.

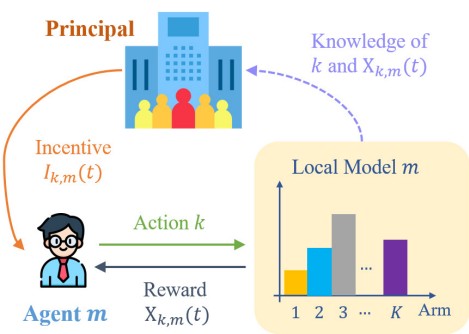

Figure 1: Incentivized exploration of the principal and agent $m$. The agent performs actions and gets both rewards and incentives. The principal pays the incentives and observes local actions and rewards.

Under the basic framework, we now formally define the learning objectives of agents and the principal.

**Agents' Objectives.** First, the self-interested agents want to collect as many rewards as possible, but note that the rewards now consist of two parts: the original rewards generated by pulled arms and additional bonuses from principal's incentives. Thus, the cumulative rewards of agent $m$ is defined as

$$R_m(T) := \sum_{t=1}^{T} \left( X_{\pi_m(t),m}(t) + I_{\pi_m(t),m}(t) \right)$$

where $\pi_m(t)$ denotes the arm pulled by agent $m$ at time $t$. We remark that such rational learning agents differ fundamentally from myopic ones assumed in most research of incentivized exploration.

**Principal' Objectives.** On the principal's side, her first goal is to identify the best global arm with a confidence higher than $1 - \delta$. Rigorously, this goal can be stated as

$$\mathbb{P}\left[ \hat{k}_*(T) = k_* \right] \geq 1 - \delta,$$

where $\hat{k}_*(T)$ denotes the identified arm at horizon $T$.

Given that the identification is correct, the principal also aims at spending as few cumulative incentives as possible, which is defined as

$$C(T) := \sum_{t=1}^{T} \sum_{m \in [M]} I_{\pi_m(t),m}(t).$$

Note that best arm identification is the principal's major task whereas minimizing cost $C(T)$ is only meaningful given $k_*(T) = k_*$ is achieved. Though not central, our mechanism will also satisfy other desirable properties such as storage efficiency.

In this work, we adopt the perspective of the principal, and try to optimize the principal's performance w.r.t. her learning objectives specified above. In other words, agents are ignorant of principal's goals and only focus on their individual rewards, and we only incorporate their "selfishness" to design an effective and efficient strategy for the principal. Such choice is natural in common applications where

the behaviors of principal (e.g., company, commercial platform) can be designed, while agents (e.g., users, customers) normally perform their own decision making which cannot be specified.

**Remark.** Note that agents and the principal actually represent two kinds of bandit learning objectives. Namely, the agents target regret minimization (Auer et al., 2002; Garivier and Cappé, 2011) (although we here use the equivalent notation of cumulative rewards instead of regret), while the principal aims at best arm identification (Audibert et al., 2010; Jamieson et al., 2014; Garivier and Kaufmann, 2016). This formulation is reasonable as agents often care more about their own cumulative benefits while the principal aims at the final result, which can then be used in the future. However, as noted in Bubeck et al. (2011), even with the same game instance, these two objectives do not necessarily align with each other, not to mention the additional global-local heterogeneity considered in our model.

## 4  A Mechanism for Incentivizing Exploration

In this section, the "Observe-then-Incentivize" (OTI) algorithm is proposed, which can effectively solve the best arm identification problem on the global model while using a small amount of incentives and maintaining efficient storage. As indicated by the name, the key idea of OTI is "Observe-then-Incentivize", which comes from its two phases: the observing phase and the incentivizing phase.

### 4.1  The "Take-or-Ban" Incentive-Provision Protocol

Before designing detailed mechanisms, we first recognize one major challenge to incentivize forward-looking agents is that even if we provide sufficient bonuses to compensate her reward at the current round, the agent may not want to take it for various reasons, e.g., giving up her current choice may lead to significant future losses, or refusing the current compensation may trick the principal to offer more future bonuses. Therefore, it is not clearly how to regulate agents' behaviors through incentives. To overcome this barrier, we propose the "Take-or-Ban" incentive-provision protocol, which provably guarantees that it is in the best interest for every agent to follow the offered incentives. This protocol is announced to the agents at the beginning of the game, and detailed as follows.

**"Take".** At time step $t$, bonuses $I_{k,m}(t)$ offered to agent $m$ are set as a binary value. Specifically, $I_{k,m}(t) = 1$ if principal wants to incentivize exploration on her arm $k$; otherwise $I_{k,m}(t) = 0$. In other words, if her arm $k$ is incentivized, agent $m$ gets reward $X'_{k,m}(t) = 1 + X_{k,m}(t)$ by pulling it.

**"Ban".** To avoid intractable agent behaviors, the following safeguard approach is adopted. Specifically, at time step $s$, if agent $m$ is provided incentive for taking some action (i.e., $\exists k, I_{k,m}(s) = 1$) but she does not pull it (i.e., $\pi_m(s) \neq k$), she is marked as "banned" by the principal. The principal stops providing bonuses for any banned agent in the future, i.e., $\forall t > s, \forall k \in [K], I_{k,m}(t) = 0$. In other words, due to her failure in following the current incentive, agent $m$ loses the chance of taking future bonuses (bu she is still free to play the local game and get original rewards $X_{k,m}(t)$).

### 4.2  The Observing Phase

Interestingly, while strategic agents introduce substantial challenges to the principal's learning problem, their fundamental need of learning local models can be a blessing. In order to collect more local rewards, agents naturally have to address the intrinsic "exploration-exploitation" dilemma. In particular, to guarantee low regrets, some (but maybe limited) explorations on each arm are required by each agent. Specifically, this argument is supported by the following asymptotic lower bound (Lai and Robbins, 1985): with any consistent agent strategy,[1] for arm $k \neq k_{*,m}$, it holds that

$$\liminf_{\Gamma \to \infty} \frac{\mathbb{E}[N_{k,m}^w(\Gamma)]}{\log(\Gamma)} \geq \frac{1}{\mathrm{KL}(\mu_{k,m}, \mu_{*,m})}, \tag{1}$$

where $N_{k,m}^w(\Gamma)$ is the number of pulls by agent $m$ on arm $k$ up to time $\Gamma$ when there are no incentives, and $\mathrm{KL}(\mu_{k,m}, \mu_{*,m})$ is the KL-divergence between two corresponding reward distributions. In other words, to guarantee local performance, agents would spontaneously explore all local arms.

On the principal's side, this observation indicates that free local information can be obtained by just letting agents directly run their own local algorithms. Thus, intuitively, it is wise not to incentivize at the beginning of the game, but to observe and enjoy the "free rides" provided by agents' exploration.

---

[1]A "consistent" strategy has a regret of $o(T^\psi)$ in any bandit instance, $\forall \psi > 0$ (Lai and Robbins, 1985).

However, since local models are often not aligned with the global model, it may not be desirable to fully rely on the "free rides" from local agents' actions since they may gradually converge to their own optimal local arms, which may not be what the principal is interested in exploring.

Thus, it is necessary to reserve some time before the end of the game for adaptive adjustments. By putting these intuitions together, in the design of OTI, we specify the observing phase to last $\kappa(T) = \frac{T}{2}$ time steps from the beginning of the game.[2] To summarize, in this observing phase, the principal does not incentivize at all, i.e., $\forall t \in [0, \frac{T}{2}], \forall k \in [K], \forall m \in [M], I_{k,m}(t) = 0$. Instead, she only observes local actions and rewards. Due to our choice of space-efficient information aggregation (specified at the end of Section 4.3), by the end of the observing phase, the principal has the record $\{N_{k,m}(\frac{T}{2}), \hat{\mu}_{k,m}(\frac{T}{2}) | \forall k \in [K], \forall m \in [M]\}$, where $N_{k,m}(t)$ is the number pulls performed by agent $m$ on arm $k$ up to time $t$ and $\hat{\mu}_{k,m}(t)$ is the corresponding sample means from these pulls. These information serve as the foundation for the remaining $\frac{T}{2}$ time slots of the incentivizing phase.

### 4.3 The Incentivizing Phase

From time slot $\frac{T}{2} + 1$, the principal enters the incentivizing phase, where she actively leverages incentives instead of passively observing. The first challenge she has is how to aggregate local

---

**Algorithm 1** OTI: Principal

---

1: Initialization: $\forall k \in [K], \forall m \in [M], N_{k,m}(0) \leftarrow 0, \hat{\mu}_{k,m}(0) \leftarrow 0$
2: **for** $t = 1, 2, \cdots, \frac{T}{2}$ **do**      ▷ *Observing Phase*
3:      $\forall k \in [M], \forall m \in [M], I_{k,m}(t) \leftarrow 0$
4:      $\forall m \in [M]$, observe $\{\pi_m(t), X_{\pi_m(t),m}(t)\}$, then update $N_{\pi_m(t),m}(t)$ and $\hat{\mu}_{\pi_m(t),m}(t)$
5: **end for**
     ▷ *Incentivizing Phase:*
6: Set $S(\frac{T}{2}) \leftarrow [K]$      ▷ *Incentivizing Phase*
7: **for** $t = \frac{T}{2} + 1, \frac{T}{2} + 2, \cdots, T$ **do**
8:      $\forall k \in [K], \hat{\mu}_k(t-1) \leftarrow \frac{1}{M} \sum_{m \in [M]} \hat{\mu}_{k,m}(t-1)$ and set $CB_k(t-1)$ with Eqn. (2)
9:      Update $S(t)$ as specified in Eqn. (3)
10:      **if** $|S(t)| \geq 1$ **then**
11:          $\bar{k}(t) \leftarrow \arg\max_{k \in S(t)} CB_k(t-1)$
12:          $\bar{m}(t) \leftarrow \arg\min_{m \in [M]} N_{\bar{k}(t),m}(t-1)$
13:          $I_{\bar{k}(t),\bar{m}(t)}(t) \leftarrow 1$
14:          $I_{k,m}(t) \leftarrow 0, \forall m \neq \bar{m}(t), \forall k \neq \bar{k}(t)$
15:      **else**
16:          $\hat{k}_*(T) \leftarrow$ the remaining arm in $S(t)$
17:      **end if**
18: **end for**
**Output:** $\hat{k}_*(T)$

---

information. Especially, the local sample means (i.e., $\hat{\mu}_{k,m}(t)$) are from different agents and associated with different number of pulls (i.e., $N_{k,m}(t)$), which further result in their individually unequal uncertainties. To address this challenge, a new (global) arm elimination algorithm is proposed, which is inspired by standard arm elimination algorithms in best arm identification (Even-Dar et al., 2002; Karnin et al., 2013) but is specifically designed to tackle the new issue of global-local heterogeneity.

Specifically, at each time step $t \in [\frac{T}{2} + 1, T]$, with the record $\{N_{k,m}(t-1), \hat{\mu}_{k,m}(t-1) | \forall k \in [K], \forall m \in [M]\}$, the principal estimates the expected reward of global arm $k$ as $\hat{\mu}_k(t-1) = \frac{1}{M} \sum_{m \in [M]} \hat{\mu}_{k,m}(t-1)$ and associates global arm $k$ with the following confidence bound:

$$CB_k(t-1) = \frac{1}{M} \sqrt{\left( \sum_{m \in [M]} \frac{1}{N_{k,m}(t-1)} \right) \left( \log\left(\frac{KT}{\delta}\right) + 4M \log\log\left(\frac{KT}{\delta}\right) \right)}. \quad (2)$$

Note that Eqn. (2) incorporates the different number of pulls on arm $k$ by *all* the local agents, i.e., $\{N_{k,m}(t-1) | \forall m \in [M]\}$. It is more challenging than the standard confidence bound design in best arm identification (Jamieson and Nowak, 2014; Gabillon et al., 2012), which only considers one source of arm pulls. This complication can be better understood in later theoretical analysis.

With the estimation and confidence bound, arms that are sub-optimal with high probabilities can be identified and eliminated while the other arms are left for more explorations in the future. Namely, let the active arm set $S(t)$ denote the arms that have not been determined to be sub-optimal up to time step $t$, which is initialized as $S(\frac{T}{2}) = [K]$, the new active arm set $S(t)$ is updated from $S(t-1)$ as

$$S(t) = \left\{ k \in S(t-1) | \hat{\mu}_k(t-1) + CB_k(t-1) \geq \max_{j \in S(t-1)} \{\hat{\mu}_j(t-1) - CB_j(t-1)\} \right\}. \quad (3)$$

To have the above-illustrated procedure effectively iterate over time, explorations are required for the arms in set $S(t)$. Hence, incentives play a critical role and the principal needs to decide which

---

[2]Note that although $\kappa(T)$ is chosen to be $\frac{T}{2}$ here, there are other possible choices, e.g., $\frac{T}{4}$, $\sqrt{T}$, etc. Details for the influence of this choice are provided in the supplementary material.

arm-agent pair to incentivize. In OTI, this decision process consists of two steps. The first step is to find the active arm with the largest confidence bound, i.e.,

$$\bar{k}(t) := \arg\max_{k \in S(t)} CB_k(t-1).$$

Then, the second step is to further identify the local agent who has the least pulls on arm $\bar{k}(t)$ i.e.,

$$\bar{m}(t) := \arg\min_{m \in [M]} N_{\bar{k}(t),m}(t-1).$$

Finally, the principal would only incentive agent $\bar{m}(t)$ to explore the arm $\bar{k}(t)$, i.e.,

$$I_{k,m}(t) \leftarrow \begin{cases} 1 & \text{if } k = \bar{k}(t) \text{ and } m = \bar{m}(t) \\ 0 & \text{otherwise} \end{cases}.$$

Intuitively, the arm-agent pair $\{\bar{k}(t), \bar{m}(t)\}$ represents the largest source of uncertainties in the current estimation of active global arms, i.e., the arm with largest confidence interval and the agent who had the fewest pulls on it. Thus, explorations on this pair is naturally the most efficient way to increase the confidence of estimations.

By iterating this process of eliminating and incentivizing, the principal would eventually have sufficient information to shrink the active arm set to have only one arm left when the horizon is sufficient, i.e., $|S(t)| = 1$, and this remaining arm is output as the identified optimal arm $\hat{k}_*(T)$.

**Space-efficient Information Aggregation.** While the principal can observe local actions and rewards, it is not storage-friendly to store all raw local data sequence $\{\pi_m(\tau), X_{\pi_m(\tau),m}(\tau) | \forall m \in [M], \forall \tau \leq t\}$, which requires memory space of order $O(Mt)$ that grows linearly in $t$. Instead, OTI is designed to keep track of $\{N_{k,m}(t), \hat{\mu}_{k,m}(t) | \forall k \in [K], \forall m \in [M]\}$. These values can be updated iteratively and only take a constant memory space of $O(KM)$ regardless of the horizon.

# 5 Theoretical Analysis

## 5.1 Main Results

The performance of OTI is theoretically analyzed from several different aspects. With agents running general consistent local strategies, the following theorem establishes the success of best arm identification and bounds the *expected* cumulative incentives.

**Theorem 1.** *It is the best interest for every agent to always accept the incentivized explorations under the "Take-or-Ban" protocol. Moreover, if the agents' local strategy is consistent without incentives and the horizon $T$ is sufficiently large, the OTI algorithm satisfies that $\mathbb{P}[\hat{k}_*(T) = k_*] \geq 1 - \delta$, and the expected cumulative incentives are bounded as*

$$\mathbb{E}[C(T)] = O\left( \sum_{k \in [K]} \sum_{m \in [M]} \left[ \frac{\log(\frac{KT}{\delta})}{M\Delta_k^2} + \frac{\log\log(\frac{KT}{\delta})}{\Delta_k^2} - \min\left\{ \frac{T}{2}, \frac{\log(\frac{T}{2})}{\mathsf{KL}(\mu_{k,m}, \mu_{*,m})} \right\} \right]^+ \right), \quad (4)$$

*where $x^+ := \max\{x, 0\}$.*

While Theorem 1 provides a general upper bound with arbitrarily consistent local strategies, Eqn. (4) is an upper bound on *expected* cumulative incentives. It does not imply a (stronger) high-probability bound. For example, the incentives may be of order $O(\log^2(T))$ with $1/\log(T)$ probability.

To better understand the performance of OTI, next we consider agents who run UCB (Auer et al., 2002), one of the most commonly adopted MAB algorithms. Specifically, agents are assumed to run the following $\alpha$-UCB algorithm (Bubeck and Cesa-Bianchi, 2012) when there are no incentives:

$$\pi_m(t) \leftarrow \arg\max_{k \in [K]} \left\{ \hat{\mu}_{k,m}(t-1) + \sqrt{\alpha \log(t)/N_{k,m}(t-1)} \right\},$$

where $\alpha$ is a positive constant specified in the design, and a typical choice is $\alpha = 2$ (Auer et al., 2002). In this case, we are able to achieve a stronger *high-probability* guarantee for incentives.

**Theorem 2.** *While the agents run $\alpha$-UCB algorithms with $\alpha \geq \frac{3}{2}$ and the horizon $T$ is sufficiently large, the OTI algorithm satisfies that $\mathbb{P}[\hat{k}_*(T) = k_*] \geq 1 - \delta$. Moreover, it holds that*

$$\mathbb{P}\left[C(T) = O\left(\sum_{k\in[K]}\sum_{m\in[M]}\left[\frac{\log(\frac{KT}{\delta})}{M\Delta_k^2} + \frac{\log\log(\frac{KT}{\delta})}{\Delta_k^2} - \frac{\alpha\log(\frac{T}{2})}{\Delta_{k,m}^2}\right]^+\right)\right] \geq 1 - \frac{4MK}{T}. \quad (5)$$

In both Eqns. (4) and (5), the first two terms represent the number of pulls that the principal needs on agent $m$'s arm $k$ to determine whether it is optimal or not. They are proportional to $1/\Delta_k^2$ as in standard best arm identification algorithms (Jamieson and Nowak, 2014). In addition, the second term is a lower-order one w.r.t. $1/\delta$. Taking a deeper look into these results, we see that the first term decreases with increasing number of agents (proportional to $1/M$). This observation indicates that increasing the population of agents actually benefits the learning of the principal. We note that the importance of *agent population* has not been fully recognized in prior studies.

Furthermore, both last terms in Eqns. (4) and (5) characterize the number of spontaneous pulls that agent $m$ performs on arm $k$ during the observing phase, which is the amount of "free pulls" taken by the principal. This term is guaranteed by Eqn. (1) for general consistent local strategies, and by the to-be presented Lemma 4 for UCB. In other words, the learning behavior of strategic agents benefits the exploration of the principal. This observation is interesting since as opposed to most prior studies of myopic agents, the principal can leverage the natural behavior of strategic agents.

Both of the aforementioned observations lead to our key result, that if there are enough amount of agents, i.e., $M$ is large, the last term dominates the first two terms, which means no incentives are needed. Correspondingly, a somewhat surprising result emerges – when there are sufficiently many learning agents involved, the exploration process of the principal can be (almost) free.

The key parts in the proofs are illustrated in the following, which may be of independent interests.

### 5.2 Proof Step 1: Effectiveness of the "Take-or-Ban" Incentive-Provision Strategy

In prior works with myopic agents, it is obvious that with sufficient instantaneous bonuses, they would pull the incentivized arms. However, this work deals with strategic agents with long-term goals. As stated in Section 4.1, these strategic agents have much more complicated behaviors, e.g., they may occasionally (instead of always) follow incentives, or they may refuse incentives first but accept later, which requires a more careful agent behavior analysis. Fortunately, our "Take-or-Ban" protocol guarantees that agents will always follow the incentives, as stated in the following lemma.

**Lemma 1.** *Under the "Take-or-Ban" incentive-provision protocol, following incentives, whenever offered, is optimal w.r.t. the expected cumulative rewards (compared to not following) for every agent.*

Note that the above lemma does not rely on agent's learning algorithm and holds for all possible ones. Thus regardless of local agents' original intentions, as long as they are self-interested and rational, *they should always follow the incentives*, i.e., pull the arm which has extra bonus offered. This design provides a clean characterization of agent behaviors for our analysis of OTI next.

**Remark.** The "Take-or-Ban" protocol is designed for theoretical rigor. For practical applications, the design OTI algorithm can be implemented with relaxed protocols; it is just that the rigorous theoretical incentive guarantee may not hold for some rational users with sophisticated strategies.

### 5.3 Proof Step 2: Effectiveness of Best Arm Identification

Some key lemmas are presented to demonstrate the effectiveness and efficiency of the designed best arm identification algorithm. Inspired by proof techniques in combinatorial MAB (CMAB) literature (Combes et al., 2015), the confidence bound design in Eqn. (2) is validated in the following lemma.

**Lemma 2.** *Denote $\mathcal{H} := \left\{\forall t \in [\frac{T}{2} + 1, T], \forall k \in [K], |\hat{\mu}_k(t-1) - \mu_k| \leq CB_k(t-1)\right\}$. When the horizon $T$ is sufficiently large, it holds that $\mathbb{P}(\mathcal{H}) \geq 1 - \delta$.*

Conditioned on event $\mathcal{H}$, the required number of local pulls is characterized in the following lemma.

**Lemma 3.** *When event $\mathcal{H}$ happens, $\forall t \in [\frac{T}{2} + 1, T]$, we have $k_* \in S(t)$, i.e., the optimal global arm would not be eliminated. Moreover, it suffices to eliminate arm $k \neq k_*$ at time $t$, i.e., $k \notin S(t)$, when*

$$\forall m \in [M], N_{k,m}(t-1), N_{k_*,m}(t-1) \geq \frac{16\log(KT/\delta)}{M\Delta_k^2} + \frac{64\log\log(KT/\delta)}{\Delta_k^2}.$$

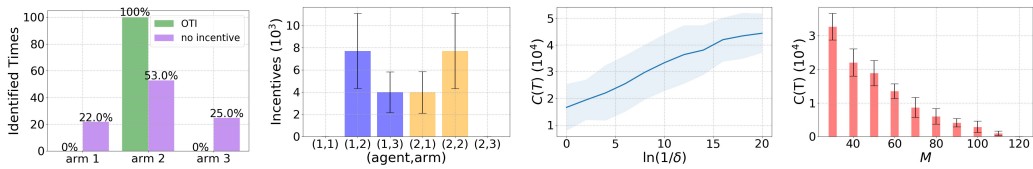

(a) With or w/o incentives. (b) Incentives assignment.      (c) Incentives v.s. $\delta$      (d) Incentives v.s. $M$

Figure 2: Experimental results. (a)-(c) are performed under a 2-agents-3-arms example while (d) is evaluated with random instances with 30 arms and varying number of agents. (a) reports the identification accuracy with and w/o incentives, (b) the assignment of incentives, (c) the logarithmic dependence on $\delta$, and (d) the diminishing effect of cumulative incentives with $M$ increasing.

With the decision process in Section 4.3 and the lower bound in Eqn. (1), Theorem 1 can be proved.

### 5.4  Proof Step 3: A Finite-time Lower Bound of UCB

An important ingredient to prove Theorem 2 is a finite-time high-probability lower bound for $\alpha$-UCB:

**Lemma 4.** *When $\Lambda$ satisfies $\frac{\Lambda}{\log^2(\Lambda)} > \frac{4K(\alpha-3/2)^2}{\Delta_{\min,m}^4}$, the $\alpha$-UCB algorithm with $\alpha \geq \frac{3}{2}$ satisfies that*

$$\mathbb{P}\left[\forall k \in [K], N_{k,m}^w(\Lambda) \geq \frac{(\sqrt{\alpha}-\sqrt{1.5})^2 \log(\frac{\Lambda}{2})}{4\Delta_{k,m}^2}\right] \geq 1 - \frac{2K}{\Lambda}. \tag{6}$$

We note that Eqn. (6) is valuable along multiple lines. First, it is a finite-time bound as opposed to the asymptotic one in Eqn. (1). Second, while Eqn. (1) holds in expectation, Eqn. (6) is a stronger high-probability bound, and implies a bound of the same order for expectation. We believe that this result itself may contribute to the understanding of UCB. Specifically, this lemma characterizes UCB's conservativeness, i.e., it would (nearly) always explore *every* arm at least logarithmic times.

## 6  Experiments

Numerical experiments have been carried out to evaluate OTI. All the results are averaged over 100 runs of horizon $T = 10^5$ and the agents perform the $\alpha$-UCB algorithm specified in Section 5.1 with $\alpha = 2$. More experimental details can be found in the supplementary material.

First, with a toy example of $M = 2$ agents and $K = 3$ arms, the ineffectiveness of not incentivizing is illustrated. Specifically, agent 1's expected rewards for the three arms are set as $[0.89, 0.47, 0.01]$ while agent 2's as $[0.01, 0.47, 0.89]$, which results in a global instance with expected rewards $[0.45, 0.47, 0.45]$.[3] Note that the optimal global arm is arm 2, while the local optimal arm is arm 1 (resp. 3) for agent 1 (resp. 2), which raises the global-local conflicts. Without incentives, the principal can only output the arm with the largest aggregated global mean at the end of the horizon. As shown in Fig. 2(a), such a "purely passive" principal does not perform the identification well. Especially, she only outputs the correct arm (i.e., arm 2) with $53\%$ accuracy. To make things worse, the principal has no control of this result, which may be even lower in other instances.

As opposed to the poor performance without incentives, Fig. 2(a) demonstrates that with incentives, OTI (using $\delta = 0.01$) can always identify the optimal global arm. Correspondingly, Fig. 2(b) presents the amount of incentives spent on each agent-arm pair. It can be observed that the principal never assigns incentives on the optimal arm of each agent, i.e., arm 1 (resp. 3) for agent 1 (resp. 2), which is intuitive since agents converge to these arms quickly and the "free pulls" on them is already sufficient. Furthermore, most of incentives are on arm 2, which is because it is sub-optimal for both agents and lacks natural explorations. Moreover, OTI is tested with varying $\delta$ under the same 2-agents-3-arms instance. Fig. 2(c) illustrates that the cumulative incentives of OTI are (nearly) proportional to $\log(1/\delta)$, which verifies the logarithmic dependence on $1/\delta$ in Eqns. (4) and (5) when $M$ is small.

At last, Fig 2(d) reports the dependence of cumulative incentives on the number of agents. Under different $M$, random local instances with 30 arms are generated to compose global instances with

---

[3]Although being a toy example, the seemingly simple instance is actually hard in terms of a small global sub-optimality gap ($\Delta_{\min} = 0.02$), large global-local divergences and a small number of involving agents.

$\Delta_{\min} \in [4.5, 5.5] \times 10^{-3}$. As shown in Fig 2(d), the cumulative incentives (with $\delta = 0.01$) gradually diminish as $M$ increases. When more than 120 agents are involved, the principal spends no incentive but can still learn the optimal global arm, which verifies our theoretical finding that with sufficiently many learning agents involved, the exploration process of the principal can be (almost) free.

# 7   Discussions and Future Works

While progresses have been made in this work, some problems are worth future investigations.

**Personalized Tasks.** This work focuses on identifying one common optimal global arm among the entire group of agents. This objective is well-motivated (Zhu et al., 2021) as only one arm can be selected for the collective interest in many applications. For example, due to the budget constraint, many companies must choose one out of multiple potential products for R&D. However, we note that it is also an interesting direction to use global information for personalized tasks, e.g., recommend items to the customer based on both her own favor and the overall popularity. The idea of FMAB with personalization in Shi et al. (2021) is one potential direction, where weighted sums with the global and local model are set as the learning objectives.

**Agent Sampling.** The principal in this work targets at learning the optimal global arm among the *involved* group of agents, which is reasonable for many use cases. For example, cellular communication operators typically can access data from all users to find the optimal channel. Moreover, to provide products to its chain stores, the company can easily perform a market survey with all of them. Nevertheless, in some applications, the principal only has access to the feedback from a sampled population but nevertheless would like to learn the best arm for the underlying entire population. We believe our findings is an important step towards this more challenging setting, where certain probably approximately correct (PAC) learning style of analysis may be needed since one may have to bound the mismatch between the best arm for the involved group and that for the whole population as the number of involved agents grows.

**Incentive-Provision Protocol.** As mentioned in Section 5.2, "Take-or-Ban" is for the purpose of rigorous theoretical analysis. Without this scheme, it is very challenging to rule out the possibility that some rational users may employ sophisticated strategies to intentionally reject an earlier incentive in order to induce higher future incentives. We note that it is intriguing to develop other agent regulation protocols or even directly analyze agents' behaviors without regulation. However, in reality, when facing less sophisticated users, we expect the insight revealed from our theoretical analysis still to be useful for less restrictive protocols, such as banning an agent for a few rounds or after a few (instead of one) refuses of the incentives (see the supplementary material for some experimental illustrations).

**Other Extensions.** In Section 5, a high-probability upper bounds of $C(T)$ is established for the agents who run $\alpha$-UCB. It is conceivable to extend the proof to other optimism-based algorithms, e.g., KL-UCB (Garivier and Cappé, 2011). However, it would be interesting to provide similar guarantees with agents running Thompson Sampling (Agrawal and Goyal, 2012) or $\epsilon$-greedy (Auer et al., 2002). Furthermore, it is worth exploring how to extend the study to other bandit types.

# 8   Conclusions

In this work, we studied incentivized exploration with multiple long-term strategic agents. Motivated by practical applications, the formulated problem involves multiple heterogeneous agents aiming at collecting high cumulative local rewards and one principal trying to identify the optimal global arm but lacking direct accesses to the global model. The OTI algorithm was designed for the principal to intelligently leverage incentives to have local agents explore information on her behalf. Three key novel components played critical roles in the design and analysis of OTI: (1) a provably effective "Take-or-Ban" incentive-provision strategy to guarantee agents' behaviors; (2) a specifically designed best arm identification algorithm to aggregate local information of varying amounts provided by heterogeneous agents; (3) a high-probability lower-bound for UCB algorithms that proved its conservativeness. The regret analysis of OTI showed that the learning behaviors of strategic agents can provide "free pulls" to benefit the principal's exploration. Moreover, we observed that increasing the population of agents can also contribute to lower the burden of principal. At last, the key and somewhat surprising result was revealed that with sufficiently many learning agents involved, the exploration process of the principal can be (almost) free.

## Acknowledgement

The CSs acknowledge the funding support by the US National Science Foundation under Grant ECCS-2029978, ECCS-2033671, and CNS-2002902, and the Bloomberg Data Science Ph.D. Fellowship. WX acknowledges the funding support by the Hong Kong Ph.D. Fellowship. HX is supported by a Google Faculty Research Award.

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
