# Supplementary Material for "(Almost) Free Incentivized Exploration from Decentralized Learning Agents"

## Chengshuai Shi, Haifeng Xu, Wei Xiong, and Cong Shen

## A    Discussions of Duration of the Observing Phase

In Section 3.3, the duration of the observing phase is specified as $\kappa(T) = \frac{T}{2}$, and we here discuss the influence of this choice and other available choices. On one hand, intuitively, if the observing phase lasts longer, more "free pulls" can be leveraged by the principal. On the other hand, there should be sufficient time reserved for adaptive adjustments, i.e., the incentivizing phase; otherwise, the principal cannot guarantee the success of best arm identification. It thus requires a careful trade-off between more "free pulls" and sufficient adaptation.

If the principal knows the parameter $\Delta_{\min}$ of the global game, i.e., the global sub-optimality gap, with Theorems 1 and 2, she can specify $\kappa(T) = \kappa_o(T) := T - \frac{16K \log(\frac{KT}{\delta})}{\Delta_{\min}^2} + \frac{64MK \log \log(\frac{KT}{\delta})}{\Delta_{\min}^2} = T - O(\log(T))$, which is an upper bound of the required number of pulls on local arms without incentives. However, it is often impossible for principal to have such information. Since best arm identification is the primal task of the principal, we choose to specify $T - \kappa(T)$ to be $\omega(\log(T))$, i.e., with an order higher than $\log(T)$, to guarantee sufficient times are left for the incentivizing phase, and the adopted $\kappa(T) = \frac{T}{2}$ is an exemplary choice among others, e.g., $\frac{T}{4}, \sqrt{T}$. As shown in the later proofs for Theorems 1 and 2, the amount of free pulls is of order $O(\log(\kappa(T)))$. Thus, while these choices ($\kappa_o(T), \frac{T}{2}, \frac{T}{4}, \sqrt{T}$, etc) seemingly distinct with each other, the amount of free pulls they provide does not differ much.

In practical applications, it is also conceivable to perform estimation of $\Delta_{\min}$ during the game with $\hat{\mu}_k(t)$, i.e., $\hat{\Delta}_{\min}(t)$, and use the estimation to determine $\kappa(T)$. However, it is difficult to provide a rigorous theoretical analysis for such an adaptive approach.

## B    Proof of Lemma 1

**Lemma 5** (Restatement of Lemma 1). *Under the "Take-or-Ban" incentive-provision protocol, following incentives, whenever offered, is optimal in terms of the expected cumulative rewards (compared to not following) for every agent.*

*Proof.* We fix an arbitrary agent $m$. Her local action $\pi_m(t)$ is made with the history $H_m(t) := \{\pi_m(\tau), X_{\pi_m(\tau),m}(\tau), I_m(\tau) | \tau \leq t-1\}$ and the current incentives $I_m(t)$, where $I_m(t) := \{I_{k,m}(t) | \forall k \in [K]\}$. Thus, we can write $\pi_m(t) = \Pi_m(H_m(t), I_m(t))$, where $\Pi_m$ is the strategy that maps the history and current incentives to actions.

Key to our proof is to argue that if strategy $\Pi_m$ does not always follow the incentives, then another strategy $\Pi'_m$ which always follows the incentives whenever offered will do better in expectation. Formally, $\Pi'_m$ is defined as follows based on a modified history $H'_m(t)$ and current incentive $I_m(t)$:

- When there is no incentive, strategy $\Pi'_m$ follows the decisions from $\Pi_m$ using the modified history $H'_m(t)$ and the observations are added to the modified history. Formally, if $\forall k \in [K], I_{k,m}(t) = 0$, then $\pi'_m(t) = \Pi'_m(H_m(t), I_m(t)) = \Pi_m(H'_m(t), I_m(t))$ and observations $\{\pi'_m(t), X_{\pi'_m(t),m}(t), I_m(t)\}$ are added to $H'_m(t+1)$.
- When there are incentives and $\Pi_m$ follows the incentive, $\Pi'_m$ also takes the incentive and adds observations to the modified history. Formally, if there exists $k \in [K]$ such that $I_{k,m}(t) = 1$ while $\Pi(H'_m(t), I_m(t)) = k$, then $\pi'_m(t) = \Pi'_m(H_m(t), I_m(t)) = k$ and observations $\{\pi'_m(t), X_{\pi'_m(t),m}(t), I_m(t)\}$ are added to $H'_m(t+1)$.
- When there are incentives but $\Pi_m$ does not take them, $\Pi'_m$ always takes the incentive, but importantly *does not add observations to the modified history*. Formally, if there exists $k \in [K]$ such that $I_{k,m}(t) = 1$ but $\Pi_m(H'_m(t), I_m(t)) \neq k$, then $\pi'_m(t) = \Pi'_m(H_m(t), I_m(t)) = k$ and no changes are made to the modified history, i.e., $H'_m(t+1) = H'_m(t)$.

If strategy $\Pi_m$ does not always take the incentives, there must be a time step $s$, $\exists k \in [K], I_{k,m}(s) = 1$ but $\Pi_m(H_m(s)) \neq k$. After time $s$, the agent is banned from taking incentives any more. The expected cumulative reward of $\pi$ can thus be decomposed as

$$\mathbb{E}[R_m^{\Pi_m}(T)] = \mathbb{E}\left[\sum_{t=1}^{s-1}(X_{\pi_m(t),m}(t) + I_{\pi_m(t),m}(t)) + \sum_{t=s}^{T} X_{\pi_m(t),m}(t)\right].$$

With strategy $\Pi'_m$, for time step $t < s$, $H'_m(t), \pi'_m(t)$ are the same as $H_m(t), \pi_m(t)$. Thus, the cumulative reward of the designed $\Pi'_m$ can also be decomposed as

$$\mathbb{E}[R_m^{\Pi'_m}(T)]$$
$$=\mathbb{E}\left[\sum_{t=1}^{T}(X_{\pi'_m(t),m}(t) + I_{\pi'_m(t),m}(t))\right]$$
$$=\mathbb{E}\left[\sum_{t=1}^{s-1}(X_{\pi'_m(t),m}(t) + I_{\pi'_m(t),m}(t)) + \sum_{t=s}^{T}(X_{\pi'_m(t),m}(t) + I_{\pi'_m(t),m}(t))\right]$$
$$=\mathbb{E}\left[\sum_{t=1}^{s-1}(X_{\pi_m(t),m}(t) + I_{\pi_m(t),m}(t)) + \sum_{t=s}^{T}(X_{\pi'_m(t),m}(t) + I_{\pi'_m(t),m}(t))\right]$$
$$\geq\mathbb{E}\left[\sum_{t=1}^{s-1}(X_{\pi_m(t),m}(t) + I_{\pi_m(t),m}(t)) + \sum_{t\in[s,T]/\tau_m^{s,T}} X_{\pi'_m(t),m}(t) + |\tau_m^{s,T}|\right]$$

where $\tau_m^{s,T}$ denotes the set of time slots that principal provides incentives in time interval $[s, T]$, i.e., $\tau_m^{s,T} = \{t \in [s,T] | \exists k \in [K], I_{k,m}(t) = 1\}$.

Since the observation from incentives are not counted in $H'_m(t)$, the distribution of $\{H'_m(t)|t \in [s,T]/\tau_m^{s,T}\}$ is the same as the distribution of $\{H_m(t)|t \in [s, T - |\tau_m^{s,T}|]\}$, which further means the distribution of $\{\pi'_m(t)|t \in [s,T]/\tau_m^{s,T}\}$ is the same with $\{\pi_m(t)|t \in [s, T - |\tau_m^{s,T}|]\}$. Thus, we can get

$$\mathbb{E}\left[\sum_{t\in[s,T]/\tau_m^{s,T}} X_{\pi'_m(t),m}(t)\right] = \mathbb{E}\left[\sum_{t\in[s,T-|\tau_m^{s,T}|]} X_{\pi_m(t),m}(t)\right].$$

With this result, it holds that

$$\mathbb{E}[R_m^{\Pi_m}(T)]$$
$$=\mathbb{E}\left[\sum_{t=1}^{s-1}(X_{\pi_m(t),m}(t) + I_{\pi_m(t),m}(t)) + \sum_{t=s}^{T} X_{\pi_m(t),m}(t)\right]$$
$$=\mathbb{E}\left[\sum_{t=1}^{s-1}(X_{\pi_m(t),m}(t) + I_{\pi_m(t),m}(t)) + \sum_{t=s}^{T-|\tau_m^{s,T}|} X_{\pi_m(t),m}(t) + \sum_{t=T-|\tau_m^{s,T}|+1}^{T} X_{\pi_m(t),m}(t)\right]$$
$$\leq\mathbb{E}\left[\sum_{t=1}^{s-1}(X_{\pi_m(t),m}(t) + I_{\pi_m(t),m}(t)) + \sum_{t\in[s,T]/\tau_m^{s,T}} X_{\pi'_m(t),m}(t) + |\tau_m^{s,T}|\right]$$
$$\leq\mathbb{E}[R_m^{\Pi'_m}(T)].$$

Thus, strategy $\Pi'_m$ always follows incentives and provides at least the same expected cumulative rewards as strategy $\Pi_m$, which does not always follow incentives. The lemma is thus proved. $\square$

## C  Proof of Lemma 2

**Lemma 6** (Restatement of Lemma 2). *Denote*
$$\mathcal{H} := \left\{\forall t \in \left[\frac{T}{2}+1, T\right], \forall k \in [K], |\hat{\mu}_k(t-1) - \mu_k| \leq CB_k(t-1)\right\}.$$

*When the horizon $T$ is sufficiently large, it holds that $\mathbb{P}(\mathcal{H}) \geq 1 - \delta$.*

*Proof.* Using the Cauchy-Shwarz inequality, we have

$$\sum_{m \in [M]} N_{k,m}(t-1) \left(\mu_{k,m} - \hat{\mu}_{k,m}(t-1)\right)^2 \leq \theta$$

$$\Rightarrow \sum_{m \in [M]} \frac{1}{N_{k,m}(t-1)} \sum_{m \in [M]} N_{k,m}(t-1)(\mu_{k,m} - \hat{\mu}_{k,m}(t-1))^2 \leq \sum_{m \in [M]} \frac{\theta}{N_{k,m}(t-1)}$$

$$\Rightarrow \left(\sum_{m \in [M]} (\mu_{k,m} - \hat{\mu}_{k,m}(t-1))\right)^2 \leq \sum_{m \in [M]} \frac{\theta}{N_{k,m}(t-1)}$$

$$\Rightarrow \frac{1}{M} \left|\sum_{m \in [M]} (\mu_{k,m} - \hat{\mu}_{k,m}(t-1))\right| \leq \frac{1}{M} \sqrt{\sum_{m \in [M]} \frac{\theta}{N_{k,m}(t-1)}}$$

$$\Rightarrow |\hat{\mu}_k(t-1) - \mu_k| \leq \frac{1}{M} \sqrt{\sum_{m \in [M]} \frac{\theta}{N_{k,m}(t-1)}}.$$

Now with the critical concentration inequality given by Lemma 7 presented in the following, and $\theta = \log\left(\frac{KT}{\delta}\right) + 4M \log\log\left(\frac{KT}{\delta}\right)$, the above implication further indicates that

$$\mathbb{P}\left[|\hat{\mu}_k(t-1) - \mu_k| \leq CB_k(t-1)\right]$$

$$\geq \mathbb{P}\left[\sum_{m \in [M]} N_{k,m}(t-1) \left(\mu_{k,m} - \hat{\mu}_{k,m}(t-1)\right)^2 \leq \theta\right]$$

$$= 1 - \mathbb{P}\left[\sum_{m \in [M]} N_{k,m}(t-1) \left(\mu_{k,m} - \hat{\mu}_{k,m}(t-1)\right)^2 \geq \theta\right]$$

$$\geq 1 - \underbrace{2e^{M+1} \left(\frac{2\left(\log\left(\frac{KT}{\delta}\right) + 4M \log\log\left(\frac{KT}{\delta}\right)\right)^2 \log\left(\frac{KT}{\delta}\right)}{M}\right)^M \cdot \frac{1}{(\log(\frac{KT}{\delta}))^{4M}}}_{:=\text{term (a)}} \cdot \frac{\delta}{KT},$$

where the last inequality is from Lemma 7 and term (a) is of order $O(\frac{M^M e^M}{\log^M(KT/\delta)})$. Thus, when $T$ is sufficiently large, $\mathbb{P}\left[|\hat{\mu}_k(t-1) - \mu_k| \leq CB_k(t-1)\right] \geq 1 - \frac{\delta}{KT}$. Finally, with a union bound over $t \in [T/2+1, T]$ and $k \in [K]$, the lemma can be proved. $\qquad\square$

**Lemma 7.** *For any $t \in [T]$, any $k \in [K]$, and any $\theta \geq M+1$, we have*

$$\mathbb{P}\left[\sum_{m \in [M]} N_{k,m}(t-1) \left(\mu_{k,m} - \hat{\mu}_{k,m}(t-1)\right)^2 \geq \theta\right] \leq 2e^{M+1}\left(\frac{(\theta-1)\lceil \theta \log(t)\rceil}{M}\right)^M e^{-\theta}.$$

*Proof.* The proof follows the ideas from Theorem 2 in Magureanu et al. (2014) and Theorem 22 in Perrault (2020). To prove Lemma 7, it suffices to prove the following two inequalities:

$$\mathbb{P}\left[\sum_{m \in [M]} N_{k,m}(t-1) \left((\mu_{k,m} - \hat{\mu}_{k,m}(t-1))^+\right)^2 \geq \frac{\theta}{2}\right] \leq e^{M+1}\left(\frac{(\theta-1)\lceil \theta \log(t)\rceil}{M}\right)^M e^{-\theta};$$

$$\tag{7}$$

$$\mathbb{P}\left[\sum_{m \in [M]} N_{k,m}(t-1) \left((\mu_{k,m} - \hat{\mu}_{k,m}(t-1))^-\right)^2 \geq \frac{\theta}{2}\right] \leq e^{M+1}\left(\frac{(\theta-1)\lceil \theta \log(t)\rceil}{M}\right)^M e^{-\theta},$$

$$\tag{8}$$

where $x^+ = \max\{x, 0\}$ and $x^- = \min\{x, 0\}$.

We first focus on proving Eqn. (7) and the same techniques can be applied to derive Eqn. (8). We fix some $\theta \geq M + 1$, and define the desired event as:

$$\mathfrak{A}(t) := \left\{ \sum_{m \in [M]} N_{k,m}(t-1) \left( (\mu_{k,m} - \hat{\mu}_{k,m}(t-1))^+ \right)^2 \geq \frac{\theta}{2} \right\},$$

and a partition of all possible pulling times as:

$$\forall \mathbf{d} \in \mathbb{N}^M, \mathfrak{B}_{\mathbf{d}}(t) := \bigcap_{m \in [M]} \left\{ \left( \frac{\theta}{\theta - 1} \right)^{d_m - 1} \leq N_{k,m}(t-1) < \left( \frac{\theta}{\theta - 1} \right)^{d_m} \right\}.$$

Since each number of pulls $N_{k,m}(t-1)$ for $m \in [M]$ is bounded by $t$, the number of possible $\mathbf{d} \in \mathbb{N}^M$ such that $\mathbb{P}(\mathfrak{B}_{\mathbf{d}}(t)) > 0$ is bounded by $\left\lceil \frac{\log(t)}{\log(\theta/(\theta-1))} \right\rceil^M$. With the following Lemma 8 and a union bound, we can get

$$\mathbb{P}(\mathfrak{A}(t)) \leq \sum_{\mathbf{d}} \mathbb{P}(\mathfrak{A}(t) \cap \mathfrak{B}_{\mathbf{d}}(t))$$

$$\leq \left\lceil \frac{\log(t)}{\log(\theta/(\theta - 1))} \right\rceil^M \left( \frac{(\theta - 1)e}{M} \right)^M e^{1-\theta}$$

$$\leq e^{M+1} \left( \frac{(\theta - 1)\lceil \theta \log(t) \rceil}{M} \right)^M e^{-\theta}$$

where the last inequality is from $\log(\frac{\theta}{\theta-1}) = \log(1 + \frac{1}{\theta-1}) \geq \frac{1/(\theta-1)}{1+1/(\theta-1)} = \frac{1}{\theta}$. $\qquad \square$

**Lemma 8.** *Let $\mathbf{d} \in \mathbb{N}^M$. Then $\mathbb{P}(\mathfrak{A}(t) \cap \mathfrak{B}_{\mathbf{d}}(t)) \leq \left( \frac{(\theta-1)e}{M} \right)^M e^{1-\theta}$.*

*Proof.* Let $\boldsymbol{\zeta} \in \mathbb{R}_+^M$. When events

$$\mathfrak{B}_{\mathbf{d}}(t) = \bigcap_{m \in [M]} \left\{ \left( \frac{\theta}{\theta - 1} \right)^{d_m - 1} \leq N_{k,m}(t-1) \leq \left( \frac{\theta}{\theta - 1} \right)^{d_m} \right\}$$

and

$$\mathfrak{A}'(t) := \bigcap_{m \in [M]} \left\{ N_{k,m}(t-1)((\mu_{k,m} - \hat{\mu}_{k,m}(t-1))^+)^2 > \frac{\zeta_m}{2} \right\},$$

happen, $\forall m \in [M]$, it holds that

$$\mu_{k,m} - \hat{\mu}_{k,m}(t-1) > \sqrt{\frac{\zeta_m}{2N_{k,m}(t-1)}} \geq \varepsilon_m := \sqrt{\frac{\zeta_m}{2(\theta/(\theta-1))^{d_m}}}.$$

Thus, the above events $\mathfrak{A}'(t)$ and $\mathfrak{B}_{\mathbf{d}}(t)$ further imply

$$\frac{\theta - 1}{\theta} \sum_{m \in [M]} \zeta_m$$

$$= \sum_{m \in [M]} 2 \left( \frac{\theta}{\theta - 1} \right)^{d_m - 1} \varepsilon_m^2$$

$$\leq \sum_{m \in [M]} 2 N_{k,m}(t-1) \varepsilon_m^2$$

$$= \sum_{m \in [M]} 4 N_{k,m}(t-1) \varepsilon_m \times \varepsilon_m - \sum_{m \in [M]} N_{k,m}(t-1) \frac{1}{8} (4 \varepsilon_m)^2$$

$$\leq \sum_{m\in[M]} 4N_{k,m}(t-1)\varepsilon_m(\mu_{k,m} - \hat{\mu}_{k,m}(t-1)) - \sum_{m\in[M]} N_{k,m}(t-1)\frac{1}{8}(4\varepsilon_m)^2$$

$$= \sum_{m\in[M]}\sum_{\tau=1}^{t-1} 4\varepsilon_m \mathbb{1}\{\pi_m(\tau)=k\}(\mu_{k,m} - X_{k,m}(\tau)) - \sum_{m\in[M]}\sum_{\tau=1}^{t-1}\frac{1}{8}(4\varepsilon_m\mathbb{1}\{\pi_m(\tau)=k\})^2$$

$$\leq \underbrace{\sum_{m\in[M]}\sum_{\tau=1}^{t-1} 4\varepsilon_m \mathbb{1}\{\pi_m(\tau)=k\}(\mu_{k,m} - X_{k,m}(\tau))}_{:=\mathfrak{C}_1(t)}$$

$$-\underbrace{\sum_{m\in[M]}\sum_{\tau=1}^{t-1}\log \mathbb{E}\left[\exp\left(4\varepsilon_m\mathbb{1}\{\pi_m(\tau)=k\}(\mu_{k,m} - X_{k,m}(\tau))\right)\right]}_{:=\mathfrak{C}_2(t)},$$

where the last inequality is because $\mu_{k,m} - X_{k,m}(\tau)$ is $\frac{1}{2}$-sub-Gaussian and it holds that

$$\mathbb{E}\left[\exp(4\varepsilon_m\mathbb{1}\{\pi_m(\tau)=k\}(\mu_{k,m} - X_{k,m}(\tau)))\right] \leq \exp\left(\frac{1}{8}(4\varepsilon_m\mathbb{1}\{\pi_m(\tau)=k\})^2\right)$$

With these results, we can further get

$$\mathbb{P}(\mathfrak{A}'(t)\cap\mathfrak{B}_{\mathbf{d}}(t)) \leq \mathbb{P}\left[\frac{\theta-1}{\theta}\sum_{m\in[M]}\zeta_m \leq \mathfrak{C}_1 - \mathfrak{C}_2\right]$$

$$\stackrel{(a)}{\leq} \exp\left(-\frac{\theta-1}{\theta}\sum_{m\in[M]}\zeta_m\right)\mathbb{E}\left[\exp\left[\mathfrak{C}_1(t) - \mathfrak{C}_2(t)\right]\right]$$

$$\stackrel{(b)}{=} \exp\left(-\frac{\theta-1}{\theta}\sum_{m\in[M]}\zeta_m\right)$$

where inequality (a) is the standard Markov inequality, and (b) is from simple algebraic multiplication.

Note that

$$\mathbb{P}(\mathfrak{A}'(t)\cap\mathfrak{B}_{\mathbf{d}}(t)) = \mathbb{P}\left[\bigcap_{m\in[M]}\left\{2\mathbb{I}\{\mathfrak{B}_d(t)\}N_{k,m}(t-1)\left((\mu_{k,m} - \hat{\mu}_{k,m}(t-1))^+\right)^2 > \zeta_m\right\}\right]$$

and

$$\mathbb{P}(\mathfrak{A}(t)\cap\mathfrak{B}_{\mathbf{d}}(t)) = \mathbb{P}\left[\sum_{m\in[M]}2\mathbb{I}\{\mathfrak{B}_d(t)\}N_{k,m}(t-1)\left((\mu_{k,m} - \hat{\mu}_{k,m}(t-1))^+\right)^2 > \theta\right].$$

Thus, with $G = M$ and $a = \frac{\theta-1}{\theta}$ in the following Lemma 9, we can finally have

$$\mathbb{P}(\mathfrak{A}(t)\cap\mathfrak{B}_k(t)) \leq \left(\frac{(\theta-1)e}{M}\right)^M e^{1-\theta}.$$

$\square$

**Lemma 9** (Lemma 8 from Magureanu et al. (2014)). *Let $G \geq 2$, $a \geq 0$. Let $\mathbf{Z} \in \mathbb{R}^G$ be a random variable such that $\forall \boldsymbol{\zeta} \in \mathbb{R}_+^G$*

$$\mathbb{P}\left[\mathbf{Z} \geq \boldsymbol{\zeta}\right] \leq \exp\left[-a\sum_{g\in[G]}\zeta_g\right].$$

*Then for $\theta \geq \frac{G}{a}$, we have*

$$\mathbb{P}\left[\sum_{g\in[G]}Z_g \geq \theta\right] \leq \left(\frac{a\theta e}{G}\right)^G e^{-a\theta}.$$

## D Proof of Lemma 3

**Lemma 10** (Restatement of Lemma 3). *When event $\mathcal{H}$ happens, $\forall t \in [\frac{T}{2} + 1, T]$, we have $k_* \in S(t)$, i.e., the optimal global arm would not be eliminated. Moreover, it suffices to eliminate arm $k \neq k_*$ at time $t$, i.e., $k \notin S(t)$, when*

$$\forall m \in [M], N_{k,m}(t-1), N_{k_*,m}(t-1) \geq \frac{16 \log(KT/\delta)}{M\Delta_k^2} + \frac{64 \log \log(KT/\delta)}{\Delta_k^2}.$$

*Proof.* When event $\mathcal{H}$ defined in Lemma 2 happens, $\forall t \in [\frac{T}{2} + 1, T], \forall k \in S(t-1)$, it holds

$$\hat{\mu}_{k_*}(t-1) + CB_{k_*}(t-1) \geq \mu_* \geq \mu_k \geq \hat{\mu}_k(t-1) - CB_k(t-1).$$

Thus, the optimal global arm would not be eliminated.

Then, as indicated in Eqn. (3), when $\hat{\mu}_{k_*}(t) - CB_{k_*}(t) \geq \hat{\mu}_k + CB_k(t)$, arm $k \neq k_*$ is ensured to be eliminated from the active arm set. Further, we note that when

$$\forall m \in [M], N_{k,m}(t-1), N_{k_*,m}(t-1) \geq \frac{16 \left( \log(\frac{KT}{\delta}) + 4M \log \log(\frac{KT}{\delta}) \right)}{M\Delta_k^2},$$

it holds that

$$\hat{\mu}_{k_*}(t-1) - CB_{k_*}(t-1) \geq \mu_{k_*} - 2CB_{k_*}(t-1) \geq \mu_{k_*} - \frac{\Delta_k}{2};$$

$$\hat{\mu}_k(t-1) + CB_k(t-1) \leq \mu_k + 2CB_k(t-1) \leq \mu_k + \frac{\Delta_k}{2},$$

which means it suffices to eliminate arm $k$. $\qquad\square$

## E Proof of Lemma 4

In this section, the proof of Lemma 4 is provided, Note that in the following proofs, we consider the standard bandit setting without incentives, i.e., the agent runs $\alpha$-UCB on her local bandit game. The proof presented here is largely inspired by Rangi et al. (2021).

**Lemma 11.** *For horizon $\Lambda$, define event*

$$\mathcal{G}_m := \left\{ \forall k \in [K], \forall t \in \left[ \frac{\Lambda}{2} + 1, \frac{3\Lambda}{4} \right], |\hat{\mu}_{k,m}(t-1) - \mu_{k,m}| \leq \sqrt{\frac{3 \log(t)}{2N_{k,m}(t-1)}} \right\}.$$

*It holds that $\mathbb{P}[\mathcal{G}_m] \geq 1 - \frac{2K}{\Lambda}$.*

*Proof.*

$$\mathbb{P}(\bar{\mathcal{G}}_m) = \mathbb{P}\left[ \exists k \in [K], \exists t \in \left[ \frac{\Lambda}{2} + 1, \frac{3\Lambda}{4} \right], |\hat{\mu}_{k,m}(t-1) - \mu_{k,m}| > \sqrt{\frac{3 \log(t)}{2N_{k,m}(t-1)}} \right]$$

$$\leq \sum_{k \in [K]} \sum_{t=\frac{\Lambda}{2}+1}^{\frac{3\Lambda}{4}} \mathbb{P}\left[ |\hat{\mu}_{k,m}(t-1) - \mu_{k,m}| > \sqrt{\frac{3 \log(t)}{2N_{k,m}(t-1)}} \right]$$

$$\leq \sum_{k \in [K]} \sum_{t=\frac{\Lambda}{2}+1}^{\frac{3\Lambda}{4}} \sum_{\tau=1}^{t-1} \mathbb{P}\left[ |\hat{\mu}_{k,m}(t-1) - \mu_{k,m}| > \sqrt{\frac{3 \log(t)}{2N_{k,m}(t-1)}}, N_{k,m}(t-1) = \tau \right]$$

$$\leq \sum_{k \in [K]} \sum_{t=\frac{\Lambda}{2}+1}^{\frac{3\Lambda}{4}} \sum_{\tau=1}^{t-1} \mathbb{P}\left[ |\hat{\mu}_{k,m}(t-1) - \mu_{k,m}| > \sqrt{\frac{3 \log(t)}{2\tau}}, N_{k,m}(t-1) = \tau \right]$$

$$\leq \sum_{k \in [K]} \sum_{t=\frac{\Lambda}{2}+1}^{\frac{3\Lambda}{4}} \sum_{\tau=1}^{t-1} 2 \exp\left( -2 \cdot \frac{3 \log(t)}{2\tau} \cdot \tau \right)$$

$$= \sum_{k \in [K]} \sum_{t=\frac{\Lambda}{2}+1}^{\frac{3\Lambda}{4}} \frac{2}{t^2}$$

$$\leq \frac{2K}{\Lambda}.$$

$\square$

**Lemma 12** (Restatement of Lemma 4). *When $\Lambda$ satisfies $\frac{\Lambda}{\log^2(\Lambda)} > \frac{4K(\alpha-3/2)^2}{\Delta_{\min,m}^4}$, the $\alpha$-UCB algorithm with $\alpha \geq \frac{3}{2}$ satisfies that*

$$\mathbb{P}\left[\forall k \in [K], N_{k,m}^w(\Lambda) \geq \frac{(\sqrt{\alpha} - \sqrt{1.5})^2 \log(\frac{\Lambda}{2})}{4\Delta_{k,m}^2}\right] \geq 1 - \frac{2K}{\Lambda}. \tag{9}$$

*Proof.* To ease the exposition, the superscript in $N_{k,m}^w(t)$ is omitted in this proof as $N_{k,m}(t)$, but note that this proof discusses the behavior of $\alpha$-UCB without incentives. For horizon $\Lambda$ satisfying $\frac{\Lambda}{\log^2(\Lambda)} > \frac{4K(\alpha-3/2)^2}{\Delta_{\min,m}^4}$, Lemma 4 indicates that event

$$\mathcal{E}_m := \{\forall k \in [K], N_{k,m}(\Lambda) \geq F_{k,m}(\Lambda)\}$$

happens with a probability at least $1 - \frac{2K}{\Lambda}$, where $F_{k,m}(\Lambda) := \frac{(\sqrt{\alpha}-\sqrt{1.5})^2 \log(\frac{\Lambda}{2})}{4\Delta_{k,m}^2}$. To prove this lemma, it suffices to prove that $\mathbb{P}[\bar{\mathcal{E}}_m] \leq \frac{2K}{\Lambda}$.

With

$$\mathcal{G}_m := \left\{\forall k \in [K], \forall t \in \left[\frac{\Lambda}{2}+1, \frac{3\Lambda}{4}\right], |\hat{\mu}_{k,m}(t-1) - \mu_{k,m}| \leq \sqrt{\frac{3\log(t)}{2N_{k,m}(t-1)}}\right\}$$

from Lemma 11, we have that

$$\mathbb{P}[\mathcal{G}_m] \geq 1 - \frac{2K}{\Lambda}.$$

Thus, it suffices to prove that with event $\mathcal{G}_m$ happening, event $\bar{\mathcal{E}}_m$ does not happen.

We prove it by contradiction. Assume that while event $\mathcal{G}_m$ happens, the event $\bar{\mathcal{E}}_m$ also happens, which means there exists arm $k$ such that $N_{k,m}(\Lambda) \leq F_{k,m}(\Lambda)$. Then, for the interval $[\frac{\Lambda}{2}+1, \frac{3\Lambda}{4}]$, we divide it into $F_{k,m}(\Lambda)$ blocks, and each block has length $\frac{\Lambda}{4F_{k,m}(\Lambda)}$. With the pigeonhole principle, there must exist one block $[t_1, t_3]$, in which arm $k$ is not pulled, i.e., $N_{k,m}(t_3) = N_{k,m}(t_1 - 1) \leq F_{k,m}(\Lambda)$.

With event $G_m$ happening, for arm $k$, it holds that $\forall t \in [t_1, t_3]$,

$$\hat{\mu}_{k,m}(t-1) + \sqrt{\frac{\alpha \log(t)}{N_{k,m}(t-1)}}$$

$$= \hat{\mu}_{k,m}(t-1) + \sqrt{\frac{3\log(t)}{2N_{k,m}(t-1)}} + \left(\sqrt{\alpha} - \sqrt{\frac{3}{2}}\right)\sqrt{\frac{\log(t)}{N_{k,m}(t-1)}}$$

$$\geq \mu_{k,m} + \left(\sqrt{\alpha} - \sqrt{\frac{3}{2}}\right)\sqrt{\frac{\log(t)}{N_{k,m}(t-1)}}$$

$$\geq \mu_{k,m} + \left(\sqrt{\alpha} - \sqrt{\frac{3}{2}}\right)\sqrt{\frac{\log(\frac{\Lambda}{2})}{F_{k,m}(\Lambda)}}$$

$$= \mu_{k,m} + 2\Delta_{k,m}$$

$$\geq \mu_{*,m} + \Delta_{k,m}.$$

We then make the following claim that

$$\forall j \in [K]/k, N_{j,m}(t_3) - N_{j,m}(t_1 - 1) \leq N_{\max} := \frac{4(\sqrt{\alpha} + \sqrt{3/2})^2 \log(\frac{3\Lambda}{4})}{\Delta_{k,m}^2}.$$

If this claim does not hold, then there exists arm $i \in [K]/k$ such that

$$N_{i,m}(t_3) - N_{i,m}(t_1 - 1) > N_{\max,m},$$

which further means there exists $t_2 \in [t_1, t_3]$ such that

$$N_{i,m}(t_2 - 1) - N_{i,m}(t_1 - 1) = N_{\max}$$

and arm $i$ is pulled at time $t_2$. For this arm $i$, at time $t_2$, with event $\mathcal{G}_m$, we have

$$\hat{\mu}_{i,m}(t_2 - 1) + \sqrt{\frac{\alpha \log(t_2)}{N_{i,m}(t_2 - 1)}}$$

$$\leq \mu_{i,m} + \sqrt{\frac{3 \log(t_2)}{2 N_{i,m}(t_2 - 1)}} + \sqrt{\frac{\alpha \log(t_2)}{N_{i,m}(t_2 - 1)}}$$

$$\leq \mu_{*,m} + \left( \sqrt{\alpha} + \sqrt{\frac{3}{2}} \right) \sqrt{\frac{\log(t_2)}{N_{i,m}(t_2 - 1)}}$$

$$\leq \mu_{*,m} + \left( \sqrt{\alpha} + \sqrt{\frac{3}{2}} \right) \sqrt{\frac{\log(\frac{3\Lambda}{4})}{N_{\max}}}$$

$$= \mu_{*,m} + \frac{\Delta_{k,m}}{2}.$$

With the property proved above for arm $k$, i.e.,

$$\hat{\mu}_{k,m}(t_2 - 1) + \sqrt{\frac{\alpha \log(t_2)}{N_{k,m}(t_2 - 1)}} \geq \mu_{*,m} + \Delta_{k,m},$$

we can observe that arm $i$ cannot be pulled at time $t_2$, which leads to a contradiction and thus proves the claim.

Since arm $k$ is not pulled in $[t_1, t_3]$, other arms must be pulled sufficiently. Using the above claim, it must hold that

$$\sum_{j \in [K]/k} N_{j,m}(t_3) - N_{j,m}(t_1 - 1) = t_3 - t_1 + 1$$

$$\Rightarrow (K - 1) N_{\max} \geq t_3 - t_1 + 1 = \frac{\Lambda}{4 F_{k,m}(\Lambda)}$$

$$\Rightarrow N_{\max} \geq \frac{\Lambda}{4 K F_{k,m}(\Lambda)}$$

$$\Rightarrow \frac{4(\sqrt{\alpha} + \sqrt{3/2})^2 \log(\frac{3\Lambda}{4})}{\Delta_{k,m}^2} = N_{\max} \geq \frac{\Lambda}{4 K F_k(\Lambda)} = \frac{\Lambda}{4K} \frac{4 \Delta_{k,m}^2}{(\sqrt{\alpha} - \sqrt{3/2})^2 \log(\frac{\Lambda}{2})}$$

$$\Rightarrow \frac{\Lambda}{\log^2(\Lambda)} \leq \frac{4K(\alpha - 3/2)^2}{\Delta_{k,m}^4} \leq \frac{4K(\alpha - 3/2)^2}{\Delta_{\min,m}^4},$$

which contradicts with the requirement for $\Lambda$ in Lemma 4. This concludes the proof. $\qquad \square$

## F  Proof of Theorem 1

**Theorem 3** (Restatement of Theorem 1). *It is the best interest for every agent to always accept the incentivized explorations under the "Take-or-Ban" protocol. Moreover, if the agents' local strategy is consistent without incentives and the horizon $T$ is sufficiently large, the OTI algorithm satisfies that $\mathbb{P}[\hat{k}_*(T) = k_*] \geq 1 - \delta$, and the expected cumulative incentives are bounded as*

$$\mathbb{E}[C(T)] = O\left( \sum_{k \in [K]} \sum_{m \in [M]} \left[ \frac{\log(\frac{KT}{\delta})}{M \Delta_k^2} + \frac{\log \log(\frac{KT}{\delta})}{\Delta_k^2} - \min \left\{ \frac{T}{2}, \frac{\log(\frac{T}{2})}{\mathsf{KL}(\mu_{k,m}, \mu_{*,m})} \right\} \right]^+ \right),$$

*where $x^+ := \max\{x, 0\}$.*

*Proof.* First, with Lemma 1, always following the incentives provides higher expected cumulative rewards than not always following. Thus, it is the best interest for every agent to always accept the incentivized explorations under the "Take-or-Ban" protocol.

As shown in Lemma 2, event $\mathcal{H}$ happens with probability at least $1 - \delta$. When event $\mathcal{H}$ happens, the optimal global arm would not be eliminated from the active arm set, which means the best arm identification succeeds as long as all other sub-optimal arms are eliminated. Thus, it suffices to analyze how many incentives are needed to eliminate all other sub-optimal arms.

Conditioned on event $\mathcal{H}$, we make the following claim regarding the cumulative incentives:

$$\forall k \in [K], \forall m \in [M], C_{k,m}(T) \leq Z_{k,m}(T) := \left[ \frac{16 \left( \log(\frac{KT}{\delta}) + 4M \log \log(\frac{KT}{\delta}) \right)}{M \Delta_k^2} - N_{k,m}^w(\frac{T}{2}) \right]^+,$$

where $C_{k,m}(T) := \sum_{t=1}^T I_{k,m}(t)$ denotes the cumulative incentives on agent $m$'s arm $k$.

To prove this claim, we first assume that there exists an arm-agent pair, namely, $(k', m')$, such that $C_{k',m'}(T) > Z_{k',m'}(T)$. We assume $k'$ is not the optimal arm $k_*$ here, but the same analysis applies for $k_*$ with minor changes. Thus, there must exist $t' \in [\frac{T}{2}+1, T]$ such that $C_{k',m'}(t'-1) = Z_{k',m'}(T)$ while $I_{k',m'}(t') = 1$. Equivalently, we have

$$N_{k',m'}(t'-1) \geq \frac{16 \left( \log(\frac{KT}{\delta}) + 4M \log \log(\frac{KT}{\delta}) \right)}{M \Delta_{k'}^2},$$

and agent $m'$ is incentivized to explore arm $k'$ at time $t'$, i.e., $\bar{k}(t') = k'$ and $\bar{m}(t') = m'$.

However, since $\bar{m}(t') = \arg \min_{m \in [M]} N_{\bar{k}(t'),m}(t'-1)$, it holds that

$$\forall m \in [M], N_{k',m}(t'-1) \geq \frac{16 \left( \log(\frac{KT}{\delta}) + 4M \log \log(\frac{KT}{\delta}) \right)}{M \Delta_{k'}^2},$$

which means $CB_{k'}(t'-1) \leq \frac{\Delta_{k'}}{4}$. Since $\bar{k}(t') = \arg \min_{k \in S(t)} CB_k(t'-1)$, it must have that

$$\forall k \in S(t-1), CB_k(t'-1) \leq \frac{\Delta_{k'}}{4}.$$

Thus, it raises a contradiction because

$$\hat{\mu}_{k_*}(t'-1) - CB_{k_*}(t'-1) \geq \mu_{k_*} - 2CB_{k_*}(t-1) \geq \mu_{k_*} - \frac{\Delta_{k'}}{2};$$

$$\hat{\mu}_{k'}(t'-1) + CB_{k'}(t'-1) \leq \mu_{k'} + 2CB_{k'}(t'-1) \leq \mu_{k'} + \frac{\Delta_{k'}}{2},$$

which means that arm $k'$ should have been eliminated and thus cannot be incentivized.

With the above claim proved, the expected cumulative incentives can be bounded as

$$\mathbb{E}[C(T)] = \mathbb{E}\left[ \sum_{k \in [K]} \sum_{m \in [M]} C_{k,m}(T) \right]$$

$$\leq \mathbb{E}\left[ \sum_{k \in [K]} \sum_{m \in [M]} \left[ \frac{16 \left( \log(\frac{KT}{\delta}) + 4M \log \log(\frac{KT}{\delta}) \right)}{M \Delta_k^2} - N_{k,m}^w(\frac{T}{2}) \right]^+ \right]$$

$$= \sum_{k \in [K]} \sum_{m \in [M]} \left[ \frac{16 \log(\frac{KT}{\delta})}{M \Delta_k^2} + \frac{64 \log \log(\frac{KT}{\delta})}{\Delta_k^2} - \mathbb{E}\left[ N_{k,m}^w(\frac{T}{2}) \right] \right]^+. \qquad (10)$$

With Eqn. (1) from Lai and Robbins (1985), if the agents' local strategies are consistent, with horizon $\Gamma$, $\forall k \neq k_{*,m}$, it holds that

$$\liminf_{\Gamma \to \infty} \frac{\mathbb{E}[N_{k,m}^w(\Gamma)]}{\log(\Gamma)} \geq \frac{1}{\mathsf{KL}(\mu_{k,m}, \mu_{*,m})},$$

which is also stated in Eqn. (1). Thus, there exists $\Gamma_0$ such that $\forall \Gamma > \Gamma_0$, it holds that $\mathbb{E}[N_{k,m}^w(\Gamma)] \geq \frac{\log(\Gamma)}{\text{KL}(\mu_{k,m}, \mu_{*,m})}$. For $k_{*,m}$, since the local strategies are consistent, $\forall \psi > 0$, $\mathbb{E}[N_{k_{*,m},m}^w(\Gamma)] \geq \Gamma - o(\Gamma^\psi)$. Thus, it holds that $\forall k \in [K], \forall \psi > 0$,

$$\mathbb{E}\left[N_{k,m}^w(\frac{T}{2})\right] = \Omega\left(\min\left\{\frac{T}{2}, \frac{\log(\frac{T}{2})}{\text{KL}(\mu_{k,m}, \mu_{*,m})}\right\}\right), \tag{11}$$

where the minimal takes care of $\text{KL}(\mu_{k,m}, \mu_{*,m}) = 0$ for arm $k_*$. By plugging Eqn. (11) into Eqn. (10), Theorem 1 is proved. $\qquad\square$

## G  Proof of Theorem 2

**Theorem 4** (Restatement of Theorem 2). *While the agents run $\alpha$-UCB algorithms with $\alpha \geq \frac{3}{2}$ and the horizon $T$ is sufficiently large, the OTI algorithm satisfies that $\mathbb{P}[\hat{k}_*(T) = k_*] \geq 1 - \delta$. Moreover, it holds that*

$$\mathbb{P}\left[C(T) = O\left(\sum_{k \in [K]} \sum_{m \in [M]} \left[\frac{\log(\frac{KT}{\delta})}{M\Delta_k^2} + \frac{\log\log(\frac{KT}{\delta})}{\Delta_k^2} - \frac{\alpha \log(\frac{T}{2})}{\Delta_{k,m}^2}\right]^+\right)\right] \geq 1 - \frac{4MK}{T}.$$

*Proof.* Theorem 2 can be proved by plugging the UCB lower bound in Lemma 4 (instead of Eqn. (1)) into the above proof of Theorem 1. $\qquad\square$

## H  Experimental Details

The codes and instructions for the experiments are publicly available at `https://github.com/ShenGroup/Observe_then_Incentivize`. The experiments are light in computation, and were all performed on a mainstream PC. A few details for the experimental setups are provided in this section. First, if there are more than one arm remaining active at horizon $T$, OTI should output the one with the largest sample mean. This approach takes care of the scenarios with an extremely small (or even zero) global sub-optimality gap. Second, we find that while the $O(\log\log(\frac{1}{\delta}))$ term in the confidence bound in Eqn. (2) is required theoretically, it is not very helpful in practice and sometimes even degrades the overall performance. Thus, in the simulation of OTI, the confidence bound is specified as $CB_k(t-1) = \frac{1}{M}\sqrt{(\sum_{m \in [M]} \frac{1}{N_{k,m}(t-1)})\log(KT/\delta)}$. It is also interesting for future works to see whether the confidence bound in Eqn. (2) can be tightened so that the $O(\log\log(\frac{1}{\delta}))$ term can be removed theoretically.

For all the simulations in Section 6, the rewards are set to follow Bernoulli distributions. Futhermore, in the experiments of Fig. 2(d), the local game instances are chosen with the following schemes to have meaningful comparisons with different number of involving agents. First, a mean vector $\nu$ with 30 arms is specified to be linearly distributed in $[0.4, 0.545]$, i.e., with gaps $0.005$. Then, for each arm $k \in [K]$, the mean $\mu_{k,m}$ of each player $m \in [M]$ is set as a sample from a truncated Gaussian distribution between 0 and 1 with mean $\nu_k$ and variance $0.01$. After this random sampling process, the local games are chosen. Then, if the corresponding global game has a sub-optimal gap $\Delta_{\min} \in [4.5, 5.5] \times 10^{-3}$, this instance is adopted; otherwise, a new instance is generated. This approach avoids the scenarios that with the number of involving

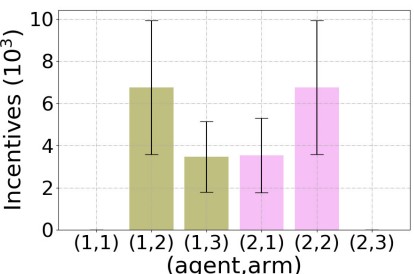

Figure 3: OTI with stochastic agent behaviors.

agents increasing, the global game becomes more and more uniform, which makes comparisons with different $M$ unfair.

Finally, additional experiments are performed to show that when dealing with relatively simple agents, OTI can be implemented with less restrictive incentive-provision protocols. In other words, the "Take-or-Ban" protocol is for theoretical rigor but may not be necessary in applications. Specifically,

the agents are set to take the incentives with probability $0.8$ and refuse with probability $0.2$. Also, the principal never bans the agent regardless of their behaviors. Using the same game instance as in Fig. 2(a), we note that with such stochastic agent behaviors, OTI can still always identify the global optimal arm correctly. The spent incentives are shown in Fig. 3, which even slightly improve the performance in Fig. 2(b). This result also illustrates the robustness of OTI.