# OpenReview forum: "(Almost) Free Incentivized Exploration from Decentralized Learning Agents"
_NeurIPS.cc/2021/Conference — NeurIPS 2021 Poster_

### Official Review · Reviewer_WEri · 2021-07-13

**Rating:** 6
**Confidence:** 3

**Summary:**

This paper studies incentivized exploration in multi-armed bandits with long-term strategic agents. The question of how to incentivize agents to explore underrated actions/arms has become an increasingly popular topic in the recent few years. Yet, it is still a relatively underexplored area for studying how to incentivize UCB agents. Therefore, I think the proposed method contributes to a significant problem.

**Limitations And Societal Impact:**

The suggestions are included together with the main review.

**Main Review:**

Below I want to summarize some of my concerns.

1. Best Arm Identification

In this paper, it is assumed that the principal aims to find the (global) best arm. I think the authors should explain more about why such a goal is significant in this setting, although three relevant papers with the same goal (best arm identification) are cited. Personally, I think "which can be used in the future" is not enough.

In the cited paper (Audibert et al., 2010), there is only one agent. In this paper, there are multiple agents with different utility functions. The heterogeneity of the agents makes the problem a little bit different. For example, if there are only two arms A and B. Half of the agents hate A and like B, while the other half like A and hate B. In such a setting, finding the "averagely" best arm seems to be useless. In recent years, "personalized incentives" and "personalized recommendations" have received more and more attention. In this sense, finding a global best arm with respect to all of the agents seems to be less attractive.

This problem is not relevant to the algorithms. What I want to stress is that I hope the authors can identify some concrete scenarios in which identifying the best global arm plays an important role.

2. Take-or-Ban

In the take-or-ban protocol, an agent will be marked as "banned" forever as long as it refuses the bonus once. I hope the authors can specify more clearly the reasons for devising such a protocol. I'm not sure what "to avoid intractable agents behaviors" means: psychologically or mathematically intractable? Also, without such a protocol, is your algorithm still able to find the best arm?

Here I understand such a protocol can simply the scenario, especially the agents' behavior. While I also want to see if it is a prerequisite for the proposed algorithm.

3. Related Works.

Some of the papers cited in Section 6 helped me a lot in understanding the motivations. Yet, the related works are summarized on the last page of the paper. I think it may be a better idea to present the related works just after the introduction part as normal because without presenting enough relevant papers, it is difficult to illustrate that the assumptions on the agents are different.



**Time Spent Reviewing:**

10

---

> ### Author Response · Authors · 2021-08-10
> **Response to Reviewer WEri**
>
> We thank the reviewer for the thoughtful suggestions, and address them in the following brief discussions.
>
> - **Best Arm Identification.**
>
>   We agree that personalized recommendation is indeed an interesting topic as well. However, we believe that our current objective of best arm identification is also equally well motivated. Though heterogeneity exists among agents, in many applications only one arm can be selected for the collective interest. For example,  due to the budget and resource constraint, many companies must choose one out of multiple potential products for R\&D and manufacturing; such decisions may rely on their market surveys which benefit from our OTI algorithm. Another example in recommender systems is when it encounters a new and unknown customer, the best recommendations could be the item with the highest ratings in the whole population. Similar formulations of targeting the averagely best arm can also be found in the study of federated MAB (Zhu et al., 2021). Thus, we believe our current problem formulation is well-defined and has practical values.
>
> - **Take-or-Ban.**
>
>   Regarding this protocol, we would like to make a few clarifications.
>
>   1. As the reviewer already noted, this protocol is for the purpose of rigorous theoretical analysis. Without it, the agents may potentially have much more uncontrollable behaviors, e.g., refusing the current incentives to mislead the principal into offering more future ones, which results in intractability of mathematical analysis. While this type of game-theoretical problems are also interesting, they are beyond the scope of the current paper. We also note that it is intriguing to develop other agent regulation protocols or even directly analyze agents' behaviors without regulation.
>   2. If we do not insist on theoretical rigor, the OTI algorithm is still functional without banning agents, i.e., "Take-or-Ban" is not a prerequisite for the developed OTI algorithm. To support this claim, we have conducted additional experiments with the same configuration as Fig. 2(a). However, in this new experiment, the agents only follow the incentives with probability $0.8$ while performing their own arm-pulling decisions with probability $0.2$, and the agents would not be banned even if they refuse incentives. This probabilistic setup provides a simple way to capture the action randomness of real-life agents. Our experimental results show that the proposed OTI algorithm can still (almost) always identify the optimal arm without paying much more incentives. We will add this set of results and discussions to the revised paper.
>
> - **Related Works.**
>
>   We agree with the kind suggestion on placing the related works after the introduction and will implement it in the final paper.

---

### Official Review · Reviewer_tYjP · 2021-07-16

**Rating:** 7
**Confidence:** 2

**Summary:**

The paper proposes incentivized exploration in multi-armed bandits (MAB) that includes multiple and long-term strategic agents that have individual learning abilities and objectives. They find that the intrinsic needs of the agents to learn benefits the principal and that increasing the number of these agents can result in (almost) free exploration process for the principal. The contribution of the paper is that it goes beyond previous researches in the field that dealt with temporary myopic agents and studied long-term strategic agents for MAB, proposing Observe-then-Incentivize (OTI) algorithm in the process.

**Limitations And Societal Impact:**

Yes.  The authors have adequately addressed the limitations and potential negative societal impact of their work.

**Main Review:**

The paper proposes incentivized exploration in multi-armed bandits (MAB) that includes multiple and long-term strategic agents that have individual learning abilities and objectives. They find that the intrinsic needs of the agents to learn benefits the principal and that increasing the number of these agents can result in (almost) free exploration process for the principal. The contribution of the paper is that it goes beyond previous researches in the field that dealt with temporary myopic agents and studied long-term strategic agents for MAB, proposing Observe-then-Incentivize (OTI) algorithm in the process.

Strengths:

The paper is clear about the contributions and the extension of previous MAB settings to incorporating long-term strategic agents has strong real-world connections and significance. It is relevant to the NeurIPS community because the study deals with decision-making process of the principal and the agents and their interactions. The Observe-then-Incentivize (OTI) algorithm is a framework that can be extended further and be applied to diverse settings where the principal interacts with agents with varying self-interests. The mathematical formulation of the setting is clear and sound.

The paper is clearly written and communicates thoroughly the connection to real-world applications. The paper is well organized in sections and is easy to follow. Also, the mathematical formulation and the setting are clearly discussed in detail and understandable.
The claims and methods are correct given their assumptions and principles. The empirical methodology is correct but would be much better if carried out with more agents and arms.


Weaknesses:

The experimental section is a little weak and could be expanded with more large-scale experiments and discussion. The “Ban” protocol is used to ensure that the agents pull the incentivized arm. According to this protocol, when an agent refuses to pull the arm that is incentivized by the principal, the principal bans that agent and does not offer any incentive in the future. This strategy seems to be limiting for the agents and breaks the high connection to real-world scenarios. More discussions need to be in place to justify this strategy.



**Time Spent Reviewing:**

3 hours

---

> ### Author Response · Authors · 2021-08-10
> **Response to Reviewer tYjP**
>
> We appreciate the reviewer's valuable feedback. Below we briefly address the two major suggestions of the reviewer.
>
>
> - **Experiments.**
>
>   We agree with the reviewer's suggestion on increasing the scale of the experiments. We note that, in the current Fig. 2(d), the number of agents $M$ already goes up to $70$, and there is no need to further increase $M$ because the cumulative incentives already approach zero. We have conducted an additional experiment with $K=30$. Similar phenomena have been observed that the cumulative incentives gradually diminish as $M$ increases, and when $M$ goes up to $110$, there are nearly no incentives that need to be paid. We will add more large-scale experiments in the revised paper, along with the corresponding discussions.
>
> - **Take-or-Ban.**
>
>   Regarding the "Take-or-Ban" protocol, we emphasize that this is designed for the purpose of theoretically and rigorously proving that all rational agents would always take the incentives when offered. Without the "ban" part, it is very challenging to rule out the possibility that some rational users may employ sophisticated strategies to intentionally reject an earlier incentive in order to induce higher future incentives.  In reality, when facing less sophisticated users, we expect the insight revealed from our theoretical analysis still to be useful for less restrictive protocols, such as banning an agent for certain number of rounds or after a few (instead of one) refuses of the incentives.
>
>   Notably, our algorithm still works with the aforementioned relaxed protocols; it is just that the rigorous theoretical incentive guarantee may not hold for some rational users with sophisticated strategies. To demonstrate this point, we have performed additional experiments with the same configuration as Fig. 2(a). However, in this new experiment, the agents are assumed to have a random behavior that mimics real-life agents: they only follow the incentives with probability $0.8$ while performing their own arm-pulling decisions with probability $0.2$. Moreover, the agents would not be banned even if they refuse incentives. Experimental results show that the proposed OTI algorithm can still (almost) always identify the optimal arm without paying much more incentives. We will add more discussions on this matter in the revised paper, and lay out possible future research on this topic.

---

### Official Review · Reviewer_jSch · 2021-07-18

**Rating:** 6
**Confidence:** 3

**Summary:**

The authors study the problem of incentivizing heterogeneous agents to explore in multi-armed bandit setting. The OTI algorithm with two phases is proposed. Theoretical guarantees are established for general agents, and for agents with UCB exploration. Moreover, a byproduct of their analysis suggests UCB's conservativeness.

**Limitations And Societal Impact:**

Yes

**Main Review:**

The writing quality and clarity are good. The idea and the approach are novel to the best of my knowledge. And the problem setting is relevant and relatively new. Below are some questions/suggestions:

(1) Why does the principal want to identify the best arm that has the highest average reward among the agents who are in the test? I hope that the authors add some discussions to this issue. For instance, if a company is only testing a product on a group of people, there may be a gap between the best arm for this group and the whole population (all the customers). Can the algorithm and the analysis be carried out under the setting where there is a ground truth for the best arm? If not, what's the technical barrier?

(2) Effect of number of agents:

(i) In the theory section, the authors highlight "proportional to $1/M$". However, there is a $\sum_{m \in [M]}$ before that term. Wouldn't this cancel the term $1/M$?

(ii) In the experiment section, $M = 60$ agents seems to be a quite large number compared to $K = 3$ arms. I think extra experiment results with larger number of arms $K$, each case run with different $M$, are desirable. With only one $K$ tested, the real dependency among $C(T)$, $K$, and $M$, seems very unclear.

Overall, the paper makes solid contribution and extension on a relevant and relatively new (but important) problem.



**Time Spent Reviewing:**

5

---

> ### Author Response · Authors · 2021-08-10
> **Response to Reviewer jSch**
>
> We thank the reviewer for the valuable suggestions. Below we address the reviewer's comments in order. These discussions will also be added to the revised paper.
>
> - **Learning objective.**
>
>   We apologize for the confusion, but want to emphasize that it is well-motivated to study the current setting of best global arm identification among the involved agents. Specifically, in many practical use cases, the participating agents are indeed the whole population, and thus there would not be a mismatch. For example, when cellular communication operators try to find the optimal channel, they typically can access data from all users. As another example, to provide products to its chain stores (agents), the company (principal) can easily perform a market survey with all of them. Moreover, this setup is already novel and non-trivial, as demonstrated in our technical sections of the paper.
>
>   Nevertheless, the situation mentioned by the reviewer is also a quite interesting setup. That is, the learner only has access to the feedback from a sampled population but nevertheless would like to learn the best arm for the underlying entire population. This is out the scope of our current study, but we believe our findings is an important step towards this more challenging setting, which we suspect certain probably approximately correct (PAC) learning style of analysis may be needed since one may have to bound the mismatch between the best arm for the involved group and that for the whole population as the number of users grows.  We will add a discussion about the reviewer's suggestion in the future work part.
>
> - **Effect of number of agents.**
>
>   1. Regarding the incentive upper bound, we apologize for the confusion, but the summing over $M$ does not cancel the $1/M$ in the first term of $C(T)$. This is because the incentive for a client $m$'s arm $k$ is not just that $1/M$ term, but depends on the difference among three additive terms applied by a $\max^+$ function (see Equations (5) and (6)). The crux here is that the third $M$-invariant term being subtracted dominates the first $1/M$ term for large $M$, which makes the $\max^+$ function equal to $0$. That is why the exploration can be almost free for the principal when $M$ is large. We will clarify this point in the revised paper.
>   2. We agree with your suggestions regarding the scale of the experiments.  Our original intention was from federated learning scenarios where a large amount of agents is one of its key characteristics.  In addition, we would like to note that in Fig. 2(d), there are $K=10$ arms tested. Experiments with a larger $K=30$  have been conducted and we observe that with $M$ more than $110$, the principal's exploration is almost free. These results will be added in to the revised paper.

---

### Decision · Program_Chairs · 2021-09-27

**Decision:**

Accept (Poster)

**Comment:**

All reviewers are positive with this submission. The reviewers agree that this paper is studying a relevant and new problem, and it is well-written. The reviewers also mention some suggestion for motivating the problem and the presentation order. I think the paper can benefit from these suggestions.